# Opioid agonist treatment and risk of death or rehospitalization following injection drug use–associated bacterial and fungal infections: A cohort study in New South Wales, Australia

**Thomas D. Brothers**[1,2,3]*, **Dan Lewer**[1,2], **Nicola Jones**[1], **Samantha Colledge-Frisby**[1], **Michael Farrell**[1], **Matthew Hickman**[4], **Duncan Webster**[3,5], **Andrew Hayward**[2], **Louisa Degenhardt**[1]

1 National Drug and Alcohol Research Centre (NDARC), UNSW Sydney, Sydney, Australia, 2 UCL Collaborative Centre for Inclusion Health, Institute of Epidemiology and Health Care, University College London, London, United Kingdom, 3 Department of Medicine, Dalhousie University, Halifax, Canada, 4 Population Health Sciences, University of Bristol, Bristol, United Kingdom, 5 Division of Infectious Diseases, Saint John Regional Hospital, Saint John, Canada

* thomas.brothers.20@ucl.ac.uk

## Abstract

### Background

Injecting-related bacterial and fungal infections are associated with significant morbidity and mortality among people who inject drugs (PWID), and they are increasing in incidence. Following hospitalization with an injecting-related infection, use of opioid agonist treatment (OAT; methadone or buprenorphine) may be associated with reduced risk of death or rehospitalization with an injecting-related infection.

### Methods and findings

Data came from the Opioid Agonist Treatment Safety (OATS) study, an administrative linkage cohort including all people in New South Wales, Australia, who accessed OAT between July 1, 2001 and June 28, 2018. Included participants survived a hospitalization with injecting-related infections (i.e., skin and soft-tissue infection, sepsis/bacteremia, endocarditis, osteomyelitis, septic arthritis, or epidural/brain abscess). Outcomes were all-cause death and rehospitalization for injecting-related infections. OAT exposure was classified as time varying by days on or off treatment, following hospital discharge. We used separate Cox proportional hazards models to assess associations between each outcome and OAT exposure. The study included 8,943 participants (mean age 39 years, standard deviation [SD] 11 years; 34% women). The most common infections during participants' index hospitalizations were skin and soft tissue (7,021; 79%), sepsis/bacteremia (1,207; 14%), and endocarditis (431; 5%). During median 6.56 years follow-up, 1,481 (17%) participants died; use of OAT was associated with lower hazard of death (adjusted hazard ratio [aHR] 0.63, 95% confidence interval [CI] 0.57 to 0.70). During median 3.41 years follow-up, 3,653 (41%) were rehospitalized for injecting-related infections; use of OAT was associated with lower hazard

**Data Availability Statement:** Requests for data access can be submitted to the National Drug and

Alcohol Research Centre (NDARC) at UNSW Sydney (ndarc@unsw.edu.au). Approval for the linkage of health data in NSW is provided under strict conditions, to protect confidentiality. Potential collaborators will be required to gain approval for data access and specific secondary analyses from the NSW Population and Health Services Research Ethics Committee. Collaborators proposing to examine research questions relating specifically to Aboriginal peoples will also be required to seek approval from the Aboriginal Health and Medical Research Council. Data may only be analysed within Australia.

**Funding:** TDB was supported by the Dalhousie University Internal Medicine Research Foundation Fellowship, Killam Postgraduate Scholarship, Ross Stewart Smith Memorial Fellowship in Medical Research and Clinician Investigator Programme Graduate Stipend (all from Dalhousie University Faculty of Medicine), a Canadian Institutes of Health Research Fellowship (CIHR-FRN# 171259), and through the Research in Addiction Medicine Scholars (RAMS) Program (National Institutes of Health/National Institute on Drug Abuse; R25DA033211) and the Fellow Immersion Training (FIT) Program in Addiction Medicine (National Institutes of Health/National Institute on Drug Abuse; R25DA013582). DL was funded by a National Institute of Health Research Doctoral Research Fellowship (DRF-2018–11-ST2-016). SC holds a Scientia PhD Scholarship from UNSW, Sydney and an Australian National Health and Medical Research Council (NHMRC) PhD Scholarship. LD is supported by an Australian National Health and Medical Research Council Senior Principal Research Fellowship (grant number 1135991). The OATS Study is supported by the National Institutes of Health (R01 DA044740 to LD). The National Drug and Alcohol Research Centre is supported by funding from the Australian Government Department of Health under the Drug and Alcohol Program. The funders had no role in study design, data collection and analysis, decision to publish, or preparation of the manuscript.

**Competing interests:** I have read the journal's policy and the authors of this manuscript have the following competing interests: In the past 3 years, LD and MF have received untied educational grant funding from Indivior and Seqirus. LD is a member of the Editorial Board of PLOS Medicine.

**Abbreviations:** aHR, adjusted hazard ratio; AMA, against medical advice; CI, confidence interval; DAG, directed acyclic graph; HIV, human immunodeficiency virus; OAT, opioid agonist treatment; OATS, Opioid Agonist Treatment Safety; PWID, people who inject drugs; RECORD-PE,

of these rehospitalizations (aHR 0.89, 95% CI 0.84 to 0.96). Study limitations include the use of routinely collected administrative data, which lacks information on other risk factors for injecting-related infections including injecting practices, injection stimulant use, housing status, and access to harm reduction services (e.g., needle exchange and supervised injecting sites); we also lacked information on OAT medication dosages.

## Conclusions

Following hospitalizations with injection drug use–associated bacterial and fungal infections, use of OAT is associated with lower risks of death and recurrent injecting-related infections among people with opioid use disorder.

---

## Author summary

### Why was this study done?

- Injecting-related bacterial and fungal infections are an increasingly common cause of pain, disability, and death among people who inject drugs (PWID).

- Treatment of injecting-related infections has tended to focus on antimicrobial therapy and/or surgery, without addressing underlying substance use-related needs.

- Opioid agonist treatment (OAT; including methadone and buprenorphine) has been associated with decreased risks of other injecting-related health harms (including HIV infection, hepatitis C virus infection, and overdose death) and may be associated with reduced risks of recurrence after injecting-related infections.

### What did the researchers do and find?

- We identified 8,943 people with opioid use disorder who were admitted to hospital with injecting-related infections in New South Wales, Australia, between 2001 and 2018.

- We found that use of OAT after hospital discharge was associated with both lower risks of death and of rehospitalization with injecting-related infections.

- Death and rehospitalization with injecting-related infections were common, even among study participants using OAT.

### What do these findings mean?

- OAT should be considered as part of a multicomponent treatment strategy for injecting-related infections, aiming to reduce death and reinfection.

REporting of studies Conducted using
Observational Routinely collected health Data
statement for PharmacoEpidemiology; SD,
standard deviation.

## Introduction

Injection drug use–associated bacterial and fungal infections (e.g., skin and soft-tissue infections, endocarditis, osteomyelitis, septic arthritis, and epidural abscess) are associated with significant morbidity and mortality among people who inject drugs (PWID) and are costly for healthcare systems [1–6]. The incidence of hospitalization for injecting-related infections is increasing in many parts of the world, including Australia [7], Canada [2,8,9], South Africa [10], the United Kingdom [11], the United States of America [12–16], and India [17].

Prevention efforts to date have focused on individual-level behavior change interventions to promote more sterile drug preparation and safer drug injecting techniques. Unfortunately, these have shown mixed results [18–20] and have had limited impact on a population level [1]. This may be in part because of social and structural factors (e.g., criminalization, discrimination, lack of access to housing, harm reduction services, and supervised injection sites) that constrain the ability of PWID to inject more safely [1,21] and that push PWID away from healthcare [22]. Improved primary and secondary prevention approaches are urgently needed [1,13,22].

One promising potential intervention to prevent injecting-related bacterial and fungal infections is opioid agonist treatment (OAT; e.g., methadone or buprenorphine). For people with opioid use disorder, OAT is associated with many benefits including reduced risks of death and of viral infections including human immunodeficiency virus (HIV) and hepatitis C virus [23,24]. OAT limits opioid withdrawal symptoms, reduces reliance on illicit drug markets, and empowers PWID to inject less frequently or in a safer way [25,26]. Engagement in OAT is also associated with regular healthcare contacts where superficial infections may be treated before they progress and become more severe or spread through the bloodstream [22,27,28].

Despite these possible benefits, in many acute care hospitals, OAT is not prioritized as part of treatment planning during and after hospitalization with injecting-related bacterial and fungal infections [22,29–31]. This is represented in low rates of OAT prescribing for these patients in multiple studies from North America [29,31,32] and in qualitative studies from the UK [22]. Suboptimal access to OAT may reflect system-level issues that separate addiction care from specialized, acute medical care for infections [1,22,29,30]. In some hospitals, clinicians have tried to overcome this by establishing specialized addiction medicine consultation services [33–36] or by infectious diseases specialists prescribing OAT directly [29,37]. While OAT is known to be beneficial for other injecting-related health outcomes, there has been relatively little research on OAT and risk for injecting-related infections. A better understanding of how OAT affects outcomes after injecting-related infections could help inform treatment planning during and following hospitalization.

Prior analyses of potential benefits of OAT after hospitalization with injecting-related infections have been limited by small sample sizes with wide confidence intervals (CIs) [38,39]. Three administrative linkage cohort studies (all from US insurance claims data) have assessed associations between use of OAT and outcomes after hospitalization with injecting-related bacterial or fungal infections [39–41]. One study identified a reduced risk of death after hospitalizations with injecting-related endocarditis, but did not assess rehospitalizations [40]. A second study identified no significant effect (with wide CIs) on risk of rehospitalization after endocarditis and did not assess mortality [39]. A third identified a reduced risk of rehospitalization for skin and soft-tissue infections at 1 year [41]. Reflecting suboptimal access, use of OAT (or of naltrexone, an opioid antagonist medication used for opioid use disorder treatment in the US) was reported as 24% within 3 months following hospital discharge in the first

study [40] and as 6% within 30 days following discharge in the second and third studies [39,41]. The latter 2 studies also only included information on buprenorphine use, as they did not have access to insurance claims or prescribing records for methadone.

The Opioid Agonist Treatment Safety (OATS) study is an administrative data linkage cohort study in New South Wales, Australia, which includes OAT permit records (with methadone or buprenorphine) for every person accessing OAT for opioid use disorder treatment in New South Wales from 2001 to 2018 [42,43].

Using data from the OATS study, we aimed to evaluate whether use of OAT, after discharge from hospital with injecting-related bacterial and fungal infections, is associated with decreased risk of subsequent mortality or infection-related rehospitalization.

## Methods

We conducted a retrospective cohort study using linked data from the OATS study, which has been described in detail elsewhere [42,43]. This manuscript follows the REporting of studies Conducted using Observational Routinely collected health Data statement for PharmacoEpidemiology (RECORD-PE) guidelines [44] (see S1 RECORD-PE checklist). Ethics approval was obtained from the NSW Population & Health Services Research Ethics Committee (2018/HRE0205), the NSW Corrective Services Ethics Committee, and the Aboriginal Health and Medical Research Council Ethics Committee (1400/18). We did not publish a protocol before conducting the analyses. The main analysis was prespecified before conducting the analyses, but the supplementary and sensitivity analyses were not prespecified.

### Setting and data sources

The OATS cohort includes all patients prescribed methadone or buprenorphine for OAT in New South Wales, which is Australia's most populous state and includes over one-third of all people receiving OAT in the country. Clinicians in NSW must apply to the state government and receive an authority to prescribe OAT for each participant. The database includes dates of OAT initiation and discontinuation. In NSW, there is no charge for OAT in public clinics or prisons. OAT may be prescribed and dispensed in specialized clinics or prescribed in primary care settings with medicine dispensed in community pharmacies.

All individuals with an OAT permit were linked to statewide hospitalization records, incarceration records, and vital statistics/death records between August 2001 and August 2018 using probabilistic linkage based on names, sex, date of birth, and Indigenous status, as described in the OATS study protocol [43].

### Participants

We included OATS study participants who survived at least one emergency (unplanned) hospitalization with skin and soft-tissue infection, sepsis or bacteremia, endocarditis, osteomyelitis, septic arthritis, or central nervous system infections (brain or spine abscess), identified using ICD-10 codes (see Fig 1 for study inclusion flow diagram; see S1 Table for ICD codes). We began with codes used in prior studies [8,29,39–41] and adapted our final list based on literature review and clinical input from the investigator team.

To be eligible, these hospitalizations had to end with the participant discharged alive to the community (rather than transfer to another hospital) so that participants could be eligible for OAT outside the hospital (see Fig 1). This was so that the timing of potential exposure and potential outcome were aligned, to avoid problems with "immortal time bias" when participants would be unable to experience either the exposure (OAT outside of acute care hospitals) or the outcomes (rehospitalization or death) [45]. Eligible hospitalizations also had to be

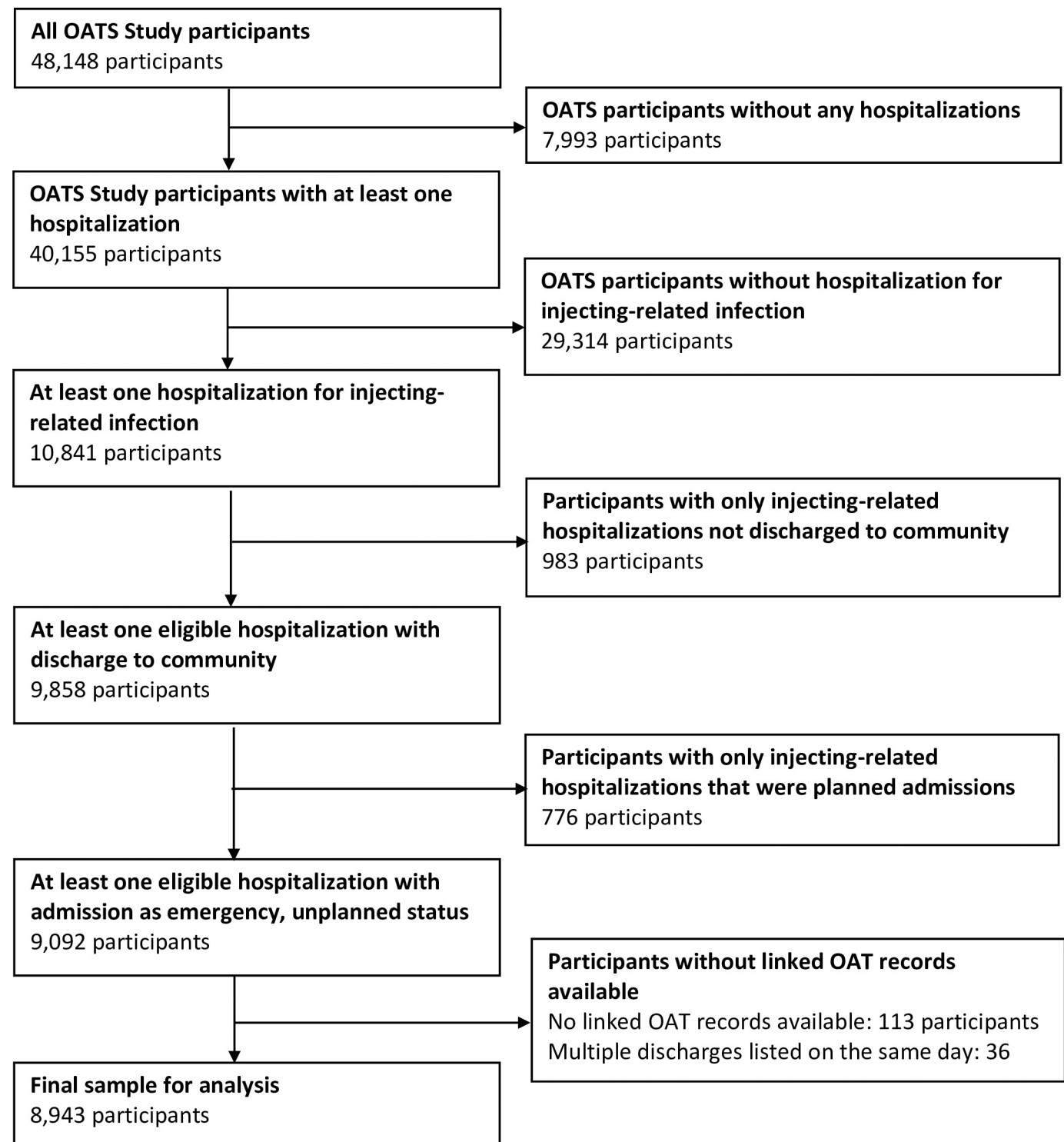

**Fig 1. Study flow diagram.** OAT, opioid agonist treatment; OATS, Opioid Agonist Treatment Safety.

emergency (unplanned) admissions. We excluded routine or planned admissions (e.g., for physical therapy or diagnostic procedures) because they are unlikely to represents episodes of acute illness attributable to injecting-related infections.

## Measures

**Outcomes.** Primary outcomes were all-cause mortality and rehospitalization with an injecting-related bacterial or fungal infection. Observed time at risk (time = 0) begins the day of discharge from participants' earliest eligible hospitalization for injecting-related infections (see Fig 2 for graphical summary of study design). Rehospitalizations for injecting-related infections were identified using the same criteria as index hospitalizations and therefore also had to be coded as emergency (unplanned) admissions. These could occur at any time point in follow-up, so may have included both hospitalizations for new infections and for failed treatments of initial infections. Participants were censored if they were still event-free on June 29, 2018.

**Inclusion criteria:**
Emergency (unplanned) hospitalization for injecting-related bacterial or fungal infection

**Exclusion criteria:**
Discharged via death or transfer to another facility

**Outcomes (Day 0 to 29 June 2018):**
All-cause death, infection-related

**Exposure window (Day 0 to 29 June 2018):**
Days on vs. off OAT (time-varying)

**Covariates (participant characteristics at index hospitalization):**
Age; Sex;
Aboriginal & Torres Strait Islander

**Covariates (participant characteristics prior to index hospitalization):**
Prior hospitalizations related to opioid, stimulant, or alcohol use;
Prior incarceration

**Covariates (characteristics of index hospitalization):**
Year of hospitalization; Length of stay;
Premature hospital discharge against medical advice;
Comorbidity

Time — Index date (hospital discharge) — End of follow-up

**Fig 2. Study design.** OAT, opioid agonist treatment.

**Primary exposure.**    The primary exposure was use of OAT, defined by dates with an active OAT prescription. OAT exposure was treated as time varying, by day of receipt. This means that each participant's follow-up time was divided into exposed (on OAT) and unexposed (off OAT) episodes (ie, medication status was not necessarily constant through follow-up) [46]. We did not stratify by type of OAT (i.e., methadone or buprenorphine) as we had no hypothesis that the protective effect would differ.

Consistent with previous studies, a new OAT episode was defined as one commencing 7 or more days after the end date of a prior treatment episode [47–50]. The same definition was used for defining the end of OAT episodes, treating the 6 days following the final day of the prescription as part of the episode. The decision to incorporate the 6 days following an OAT episode into the exposure definition was originally based on consultation with clinicians and pharmacologists [50]; it has been used in previous studies by members of our group [50,51], and similar cutoffs (e.g., 3 to 6 days) have been used by others [52,53]. This approach may introduce bias by allocating outcomes to the treatment period when they actually occurred after leaving treatment; this may overestimate rates of outcomes in-treatment (on OAT) and underestimate rates of outcomes out-of-treatment (off OAT), resulting in conservate estimates of potential benefit.

**Covariates.**    See S1 Fig for a directed acyclic graph (DAG) describing the hypothesized relationships between OAT status, the outcomes of interest, and potential confounders. All covariates were extracted from linked hospital administrative records, unless otherwise specified.

Participant characteristics measured at the time of index hospitalization included age in years (centered to mean and standardized to units of 1 standard deviation [SD]), sex (female or not female), Indigenous status (identification as Aboriginal/Torres Strait Islander or not Indigenous), and comorbidity (defined by the count of unique ICD-10 chapters recorded in any diagnostic position for the index admission). Participant characteristics measured prior to the index hospitalization (all treated as binary) include any prior acute care hospitalizations related to poisoning or toxicity from opioids (as indicators of addiction severity; T40.0 to T40.6), alcohol (F10.0, X45, X65, Y15, T51.0), or stimulants (T40.5 T43.6), and a history of prior incarceration (which is associated with increased risk for unsafe injection practices). Dates of incarceration were derived from linked incarceration administrative records.

Characteristics of the index hospitalization include the year of admission (grouped as 2001 to 2006, 2007 to 2011, or 2012 to 2018), length of stay in days (as an indicator of initial illness severity; centered to mean and standardized to units of 1 SD), and premature patient-initiated discharge against medical advice (AMA; treated as binary). For descriptive purposes, we also classified hospitalizations by the presence of each type of injecting-related infection.

## Analysis

All analyses were conducted using R version 3.6.3. We calculated the incidence rate (with Poisson CIs) of each outcome per person-time while exposed to OAT and per person-time while unexposed to OAT during follow-up. We then described the cumulative hazard of each outcome, by OAT exposure periods, using Kaplan–Meier curves and the Simon–Makuch extension for time-varying exposures [54]. These can be interpreted as the estimated survival for patients who did not change their OAT status during follow-up. We then used Cox proportional hazards models to estimate the association between OAT and the study outcomes to generate hazard ratios, adjusting for all covariates.

**Supplementary analyses.**    The relationship between OAT use and the outcomes (mortality or rehospitalization with injecting-related infection) may vary over time and OAT may have a

larger effect closer to the time of initial hospital discharge. As such, we performed a post hoc (not prespecified) supplementary analysis to generate period-specific hazard ratios within the first year after hospital discharge, within years 2 to 3, and within years 4 to 6. We did this as an extension of our final multivariable models in the main survival analyses, adjusting for all prespecified covariates.

**Sensitivity analyses.** We conducted several post hoc sensitivity analyses to test the robustness of our main analysis. First, we tested the impact of alternative OAT exposure period definitions. In our main analysis (described above), we prespecified that the 6 days following the end of an OAT episode is counted as part of the exposure. We tested whether we found similar results when reducing this exposure period to the 2 days following the OAT episode and when extending it to 10 days following the OAT episode.

We then conducted a sensitivity analysis to address a potential source of "immortal time bias" in the mortality outcome survival analysis. Immortal time occurs when, within an observation period, there is a period of time where an outcome event cannot possibly have occurred [45,55]. Because linkage between OAT record data and hospitalization data was retrospective, some participants may have had their initial hospitalization before their initial OAT record and would have been unable to experience death during this time (in other words, the fact that they have a future OAT record means they could not have died before then). We therefore constructed a new analytic sample only among participants who experienced hospitalization for injecting-related infection after their first record of OAT. We did not feel this potential issue with immortal time bias would affect the rehospitalization outcome survival analysis because participants could have experienced a rehospitalization event at any time (in this case, the fact that they have a future OAT record does not necessarily mean they could not have been hospitalized before then).

## Results

### Participants

We identified 8,943 participants with at least 1 hospitalization for injecting-related bacterial or fungal infections. Characteristics of the sample are summarized in Table 1. Participants were mostly men (66.0%), and median age at study entry was 38 years. Skin and soft tissue infections were present during most hospitalizations (see Table 1), and 14% of participants experienced a premature discharge "against medical advice." Length of stay had a right-skewed distribution, with median 4 days, 75th percentile 8 days, and 99th percentile 65 days.

Just under half of participants (4,292; 48%) were receiving OAT at the time of discharge from their index hospitalization for injecting-related infections. Of 4,651 (52%) participants without an active OAT prescription at the time of their index hospitalization, most did not access OAT soon after discharge. For example, 199 (4%) participants initiated OAT within 1 week of hospital discharge, 410 (9%) participants initiated OAT within 4 weeks, and 706 (15%) within 12 weeks.

### Main results

**All-cause mortality.** Out of 8,943 participants, 1,481 (17%) died during follow-up. In total, participants were followed for 65,240 person-years (median 6.56 years of follow-up per person), including 34,146 (52%) person-years exposed to OAT and 31,094 (48%) person-years unexposed. Of all participants, 2,174 (24%) remained exposed to OAT throughout the entire follow-up period, and 1,341 (15%) remained unexposed throughout.

Of the deaths, 643 (43%) occurred during an OAT exposure period, and 838 (57%) occurred while unexposed to OAT. Mortality rates were 1.88 deaths (95% CI 1.17 to 2.03) per

**Table 1. Descriptive characteristics of the sample.**

| Variable | Levels | Total (100%) |
|---|---|---|
| **Sample** | **N (%)** | **8,943 (100%)** |
| **Participant characteristics** | | |
| Age | Mean ± SD | 39 ± 11 |
| | Median [IQR] | 38 [31 to 46] |
| Sex | Female | 3,080 (34%) |
| | Male | 5,863 (66%) |
| Aboriginal or Torres Strait Islander | Yes | 1,321 (15%) |
| | No | 7,554 (85%) |
| | Unknown | 66 (<1%) |
| Comorbidities[1] | Median [IQR] | 3 [2 to 5] |
| | 1 | 1,183 (13%) |
| | 2 | 1,620 (18%) |
| | 3 | 1,825 (20%) |
| | 4 | 1,418 (16%) |
| | 5 | 1,040 (12%) |
| | 6+ | 1,857 (21%) |
| Prior opioid-related hospitalization | Yes | 749 (8%) |
| | No | 8,194 (92%) |
| Prior stimulant use-related hospitalization | Yes | 205 (2%) |
| | No | 8,738 (98%) |
| Prior alcohol use-related hospitalization | Yes | 929 (10%) |
| | No | 8,014 (90%) |
| Prior experience of incarceration | Yes | 3,845 (43%) |
| | No | 5,098 (57%) |
| **Index hospitalization characteristics** | | |
| Year of hospitalization | 2001 to 2006 | 2,772 (30%) |
| | 2007 to 2011 | 2,412 (27%) |
| | 2012 to 2018 | 3,809 (43%) |
| Distribution of infections[2] | Total | 8,943 (100%) |
| | Skin and soft tissue | 7,021 (79%) |
| | Sepsis/bacteremia | 1,207 (14%) |
| | Endocarditis | 431 (5%) |
| | Osteomyelitis | 375 (4%) |
| | Septic arthritis | 323 (4%) |
| | Central nervous system | 69 (1%) |
| OAT prescription active at time of discharge | Yes | 4,292 (48%) |
| | No | 4,651 (52%) |
| Length of stay (days) | Mean ± SD | 8.9 ± 42 |
| | Median [IQR] | 4 [2 to 8] |
| Discharge against medical advice | Yes | 1,246 (14%) |
| | No | 7,697 (86%) |

[1]Comorbidities defined by the number of ICD-10 chapters listed during the index hospital admission.

[2]Percentages sum to greater than 100% because each hospitalization may have codes for multiple infection categories.

AMA, against medical advice; IQR, interquartile range; SD, standard deviation; OAT, opioid agonist treatment.

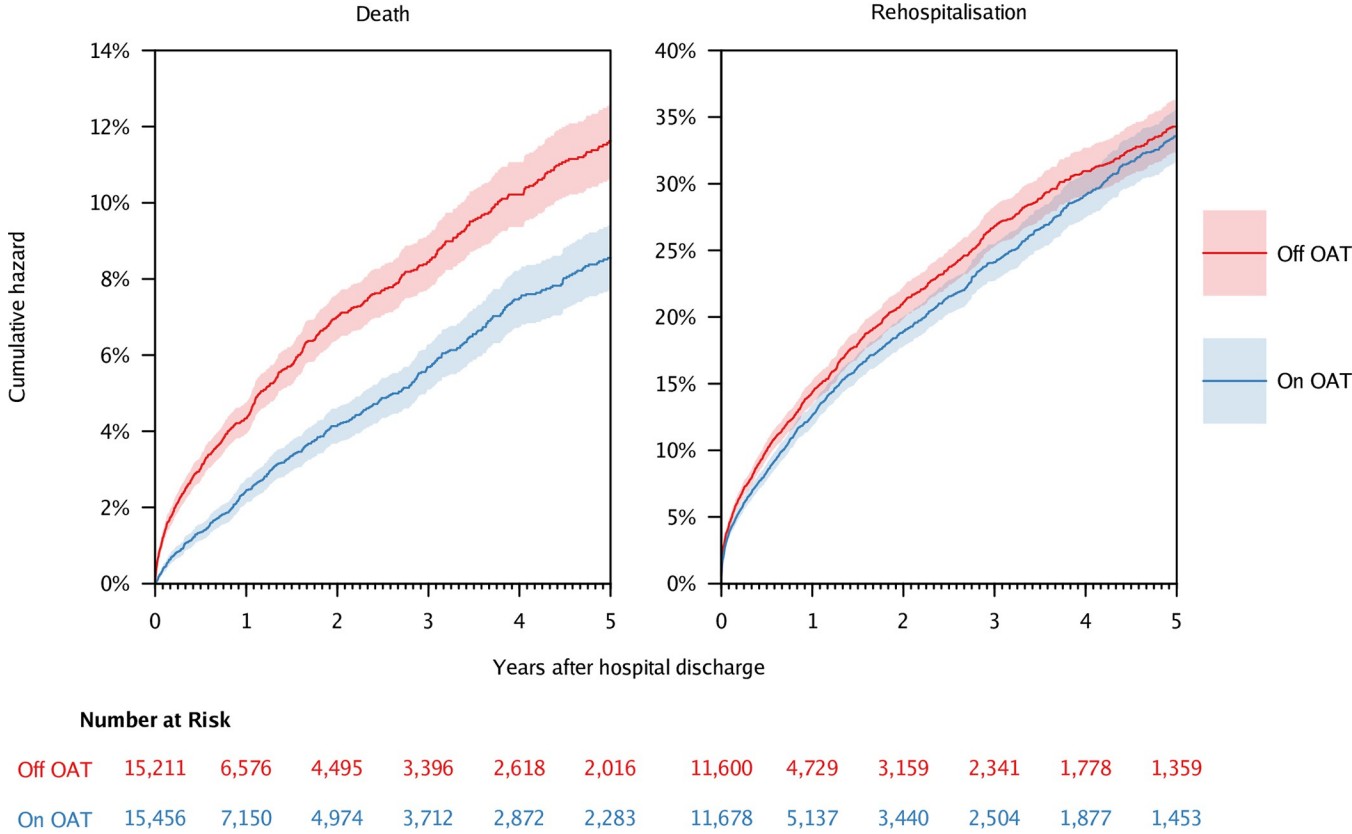

**Fig 3. Extended Kaplan–Meier curves for time to death and time to rehospitalization among participants in the OATS study who survived an initial hospitalization with injecting-related bacterial or fungal infection.** Both analyses involve 8,943 participants. The death analysis was based on 30,667 treatment or nontreatment periods, and the rehospitalization analysis was based on 23,278 treatment or nontreatment periods. OAT, opioid agonist treatment; OATS, Opioid Agonist Treatment Safety.

100 person-years exposed to OAT and 2.69 (2.51 to 2.88) per 100 person-years unexposed to OAT.

Extended Kaplan–Meier survival curves for time to death are presented in Fig 3. Cumulative hazard for death in OAT treatment versus nontreatment periods was 0.3% versus 1.2% at 30 days, 0.8% versus 2.1% at 90 days, and 2.4% versus 4.3% at 365 days.

Results of survival models are presented in Table 2. In the adjusted model, OAT was associated with lower hazard of all-cause death (adjusted hazard ratio [aHR] 0.63, 95% CI 0.57 to 0.70).

**Rehospitalization for an injecting-related infection.** Out of 8,943 participants, 3,653 (41%) were rehospitalized with an injecting-related bacterial or fungal infection. The distribution of infection type for these rehospitalizations was similar to the distribution during the index hospitalization. This included 2,718 (78%) hospitalizations with skin and soft-tissue infections, 556 (15%) with sepsis, 255 (7%) with endocarditis, 254 (7%) with osteomyelitis, 144 (4%) with septic arthritis, and 53 (1%) with central nervous system infections.

Participants were followed for 44,690 person-years (median 3.41 years per participant), which included 22,987 (51%) person-years exposed to OAT and 21,703 (49%) person-years unexposed. Of all 8,943 participants, 2,693 (30%) remained exposed to OAT throughout the entire follow-up period, and 2,157 (24%) remained unexposed throughout.

Of the rehospitalizations, 1,820 (50%) occurred during an OAT exposure period, and 1,833 (50%) occurred while unexposed to OAT. Incidence rates for rehospitalization with injecting-

**Table 2. Results of Cox regression for survival following discharge from index hospitalization with an injecting-related bacterial or fungal infection.**

| Variable | Levels | Mortality outcome | | Rehospitalization outcome[1] | |
|---|---|---|---|---|---|
| | | Unadjusted hazard ratio (95% CI) | aHR (95% CI)[2] | Unadjusted hazard ratio (95% CI) | aHR (95% CI)[2] |
| **Primary exposure** | | | | | |
| OAT | Unexposed day | Ref | Ref | Ref | Ref |
| | Exposed day | 0.72 (0.64 to 0.79) | 0.63 (0.57 to 0.70) | 0.95 (0.89 to 1.01) | 0.89 (0.84 to 0.96) |
| **Participant characteristics** | | | | | |
| Age | Years (scaled) | 2.15 (2.04 to 2.26) | 2.04 (1.93 to 2.17) | 1.33 (1.29 to 1.37) | 1.26 (1.22 to 1.31) |
| Sex | Male | Ref | Ref | Ref | Ref |
| | Female | 0.83 (0.74 to 0.92) | 0.92 (0.82 to 1.02) | 1.05 (0.99 to 1.13) | 1.09 (1.02 to 1.17) |
| Aboriginal or Torres Strait Islander | No | Ref | Ref | Ref | Ref |
| | Yes | 0.72 (0.61 to 0.85) | 1.02 (0.86 to 1.20) | 0.95 (0.86 to 1.04) | 1.00 (0.91 to 1.10) |
| | Unknown | 0.92 (0.52 to 1.62) | 0.95 (0.54 to 1.69) | 0.57 (0.37 to 0.88) | 0.62 (0.41 to 0.96) |
| Comorbidities | 1 | Ref | Ref | Ref | Ref |
| | 2 | 1.46 (1.14 to 1.89) | 1.39 (1.09 to 1.78) | 1.14 (1.01 to 1.28) | 1.09 (0.97 to 1.23) |
| | 3 | 1.88 (1.49 to 2.38) | 1.74 (1.38 to 2.20) | 1.15 (1.02 to 1.29) | 1.10 (0.98 to 1.24) |
| | 4 | 2.19 (1.73 to 2.79) | 1.98 (1.55 to 2.51) | 1.29 (1.14 to 1.46) | 1.20 (1.06 to 1.36) |
| | 5 | 3.18 (2.50 to 4.05) | 2.58 (2.03 to 3.30) | 1.54 (1.35 to 1.75) | 1.34 (1.18 to 1.54) |
| | 6+ | 5.09 (4.09 to 6.34) | 3.49 (2.79 to 4.36) | 1.83 (1.63 to 2.06) | 1.49 (1.32 to 1.68) |
| Prior opioid-related hospitalization | No | Ref | Ref | Ref | Ref |
| | Yes | 1.15 (1.02 to 1.30) | 1.12 (0.98 to 1.28) | 1.33 (1.18 to 1.49) | 1.11 (0.98 to 1.25) |
| Prior stimulant use-related hospitalization | No | Ref | Ref | Ref | Ref |
| | Yes | 0.83 (0.66 to 1.06) | 1.05 (0.82 to 1.34) | 1.20 (0.96 to 1.49) | 1.07 (0.85 to 1.34) |
| Prior alcohol use-related hospitalization | No | Ref | Ref | Ref | Ref |
| | Yes | 1.09 (0.96 to 1.24) | 1.06 (0.93 to 1.21) | 1.31 (1.18 to 1.46) | 1.16 (1.04 to 1.30) |
| Prior experience of incarceration | No | Ref | Ref | Ref | Ref |
| | Yes | 0.76 (0.68 to 0.84) | 1.00 (0.89 to 1.12) | 0.99 (0.93 to 1.06) | 1.02 (0.96 to 1.10) |
| **Index hospitalization characteristics** | | | | | |
| Era of hospitalization | 2001 to 2006 | Ref | Ref | Ref | Ref |
| | 2007 to 2011 | 1.25 (1.11 to 1.41) | 0.94 (0.83 to 1.07) | 1.13 (1.04 to 1.23) | 1.02 (0.94 to 1.11) |
| | 2012 to 2018 | 1.64 (1.44 to 1.87) | 0.83 (0.72 to 0.96) | 1.73 (1.60 to 1.87) | 1.33 (1.22 to 1.46) |
| Length of stay | Days (scaled) | 1.04 (1.02 to 1.06) | 1.02 (0.99 to 1.06) | 1.03 (1.01 to 1.04) | 1.01 (0.99 to 1.04) |

(*Continued*)

**Table 2.** (Continued)

| Variable | Levels | Mortality outcome | | Rehospitalization outcome[1] | |
|---|---|---|---|---|---|
| | | Unadjusted hazard ratio (95% CI) | aHR (95% CI)[2] | Unadjusted hazard ratio (95% CI) | aHR (95% CI)[2] |
| Discharge against medical advice | No | Ref | Ref | Ref | Ref |
| | Yes | 0.94 (0.81 to 1.10) | 1.10 (0.94 to 1.28) | 1.41 (1.30 to 1.54) | 1.47 (1.34 to 1.60) |

[1]Rehospitalization with injecting-related infection.

[2]Fully adjusted model includes all variables listed in the table.

aHR, adjusted hazard ratio; AMA, against medical advice; CI, confidence interval; OAT, opioid agonist treatment.

related infection were 7.92 (95% CI 7.66 to 8.29) per 100 person-years exposed to OAT, and 8.45 (8.06 to 8.84) per 100 person-years unexposed to OAT.

Extended Kaplan–Meier survival curves for time to rehospitalization are presented in Fig 3. Cumulative hazard for rehospitalization in OAT treatment versus nontreatment periods was 3.7% versus 4.3% at 30 days, 6.0% versus 7.1% at 90 days, and 12.7% versus 14.4% at 365 days.

In the adjusted model, OAT was also associated with lower hazard of rehospitalization (aHR 0.89, 95% CI 0.84 to 0.96; Table 2).

## Other analyses

**Supplementary analyses.** In a post hoc supplementary analysis, we explored associations between OAT and mortality or rehospitalization for injecting-related infections at different points in follow-up using period-specific hazard ratios (Table 3).

**Sensitivity analyses.** We conducted post hoc sensitivity analyses exploring the impact of alternative OAT exposure timing definitions. Changing our exposure definition to incorporate the 2 days following the end of the OAT episode (reduced from 6 days in the main analysis) demonstrated similar results for the association between OAT with all-cause mortality (aHR 0.51, 95% CI 0.46 to 0.57) and with rehospitalization (aHR 0.88, 95% CI 0.83 to 0.95). Extending the exposure period to incorporate 10 days following the end of the OAT episode also demonstrated similar results for mortality (aHR 0.72, 95% CI 0.65 to 0.80) and for rehospitalization (0.89, 95% CI 0.84 to 0.95).

We then conducted a post hoc sensitivity analysis for the mortality outcome, reconstructing the analytic sample only among participants who experienced hospitalization for injecting-related infection at a date following their first record of OAT. This sample was slightly smaller (n = 7,641). Compared to the main analysis, more participants (59%) had an active OAT permit at the time of discharge from their index hospitalization, and more follow-up time was exposed to OAT (59%). In the fully adjusted model in this smaller sample, OAT was also associated with lower hazard of all-cause death (aHR 0.56, 95% CI 0.51 to 0.62).

**Table 3.** Period-specific aHRs for associations between OAT and all-cause mortality or rehospitalization for injecting-related infections.

| Time since hospital discharge | Mortality outcome | Rehospitalization outcome |
|---|---|---|
| Within first year | 0.47 (0.40 to 0.55) | 0.83 (0.77 to 0.91) |
| Year 2 to 3 | 0.66 (0.54 to 0.81) | 0.87 (0.76 to 0.99) |
| Year 4 to 6 | 0.76 (0.58 to 0.98) | 1.10 (0.91 to 1.33) |

Hazard ratios (with 95% CIs) are for OAT exposure in fully adjusted models for all covariates.

aHR, adjusted hazard ratio; CI, confidence interval; OAT, opioid agonist treatment.

## Discussion

Among a large cohort of people with opioid use disorder who have been hospitalized with injecting-related bacterial or fungal infections, we found that OAT was associated with lower risk of mortality and of rehospitalization with these infections. Our findings of an association between OAT and lower risk of death among people with opioid use disorder are consistent with prior evidence. The magnitude of the association between OAT and lower rehospitalization risk was more modest, but we are not aware of other interventions shown to reduce risk of reinfection in this setting. Rates of death and rehospitalization remained high for this young cohort of patients, even among those exposed to OAT. Half of the sample were not prescribed OAT at the time of discharge from their initial infection-related hospitalization, and only 15% of these participants initiated OAT in the 3 months following. This suggests that OAT should be offered as part of a multicomponent treatment strategy for injecting-related infections, aiming to reduce death and reinfection.

Our findings on the benefits of OAT engagement for patients after injecting-related infection in Australia build on mixed evidence from US insurance claims databases with lower rates of OAT exposure and smaller sample sizes. One previous study, among patients with injecting-related infective endocarditis in Massachusetts, US, showed time-varying exposure to OAT or extended-release naltrexone (an opioid antagonist) after hospitalization was associated with reduced risk of death [40]. A study of patients with injecting-related infective endocarditis in a US nationwide commercial insurance claims database examined associations between buprenorphine or naltrexone within 30 days after hospital discharge and risk of rehospitalization; effect estimates were associated with wide CIs that could include both beneficial or harmful effects [39]. The sample was smaller than ours (768 participants), and less than 6% of patients were exposed to these medications during follow-up [39]. In another study analyzing patients with injecting-related skin and soft tissue infections in the same US insurance claims database, 5.5% were exposed to buprenorphine or naltrexone in 30 days following hospital discharge, and this was associated with lower risk of rehospitalization with skin and soft tissue infections at 1 year of follow-up [41]. In a retrospective chart review study of patients admitted to a Missouri, US, hospital with injecting-related bacterial or fungal infections, those who received OAT during their hospitalization and continued it at discharge were less likely to be readmitted for injecting-related infections [56]. Our findings offer more robust supportive evidence of the beneficial effects of OAT exposure following hospitalization with multiple types of injecting-related infections, a larger sample size, and higher rates of OAT exposure with more specific effect estimates.

In the present study, we identified larger effect estimates for associations between OAT use and mortality than for associations between OAT use and rehospitalization with injecting-related infections. Our findings of a large protective effect of OAT on mortality risk reduction are in keeping with prior research, including multiple observational studies showing protective effects on all-cause mortality, opioid overdose deaths, and multiple other specific causes of death (including suicide, cancer, alcohol related, and cardiovascular related) [23,57]. Future research should investigate associations between OAT and specific causes of death after hospitalization with injecting-related infections. We hypothesized several pathways through which OAT might reduce risks of recurrence of injecting-related infections, including reducing frequency of opioid injecting, improving healthcare contacts, and reducing the impacts of criminalization and violence, but we were unable to explore specific mechanisms in this study of administrative data [1,26]. People accessing OAT may still be at risk of injecting-related infections through several pathways, including ongoing injection opioid use while on OAT, suboptimal access to safe housing and harm reduction services (e.g., needle exchange and supervised

consumptions sites) and by injecting stimulants. OAT is known to reduce risks of death even among people who continue to use nonmedical or criminalized opioids [58], who may still be at risk of injecting-related infections. More research is needed to understand how to further reduce risks of injecting-related infections for people both on and off OAT.

Despite the known benefits of OAT for mortality risk reduction, less than half of participants in our study had an active prescription for OAT at the time of discharge from their index hospitalization with injecting-related bacterial or fungal infections. Published rates of OAT engagement as part of discharge planning following hospitalization with injecting-related infections vary widely, including 8% in Boston, Massachusetts, US [31] and 81% in Saint John, New Brunswick, Canada [29]. Improving access to OAT requires clinical and regulatory changes, including improved education for health professionals, increasing the number of points of access and availability on-demand, facilitating multiple medication options, and decreasing out-of-pocket patient costs [59]. Infectious disease specialists should consider integrating OAT into their care of patients with injecting-related infections [29,60]. Addiction medicine physicians can be incorporated into multidisciplinary teams to help care planning for these patients [30]. The time period immediately following discharge from acute care hospitalization is a particularly dangerous time for people with opioid use disorder [61], and so hospital-based healthcare providers should offer OAT initiation and facilitate a seamless transition to ongoing, outpatient care [27,29,33,56]. Risks of death and rehospitalization remain high among people with opioid use disorder even when engaged in OAT. Addiction treatment should be considered as part of a multicomponent secondary prevention strategy that could include consideration of environmental determinants like housing and access to other harm reduction services [1,62].

Our study has some important limitations. First, the OATS study cohort does not include all people who inject opioids in NSW; only those who have accessed OAT at least once during the study period are eligible for linkage and inclusion. However, this has previously been estimated to include >75% of people with opioid use disorder in NSW [28] and, to our knowledge, our study includes the largest sample to date of people with opioid use disorder following hospitalization with injecting-related infections. Second, as this is a study of administrative healthcare data, we have no information on additional factors that may influence risk for these infections, including individual injecting practices, housing status, and access to needle exchange or supervised injection sites [1]. We had only limited information on other social determinants, aside from prior incarceration (reflecting experiences of criminalization and possible unsafe injecting technique while incarcerated) and Aboriginal or Torres Strait Islander status (reflecting experiences of cultural strengths as well as settler colonialism and structural racism) [1]. These covariates were treated as time fixed at baseline (i.e., not time varying); further research is needed to understand whether social exposures like incarceration have time-dependent effects on injecting-related infections. Third, we did not have reliable information on the dose received each day, so did not include OAT dosing information. Fourth, oral methadone and sublingual buprenorphine were the only OAT medications used in NSW during the study period, so we were unable to estimate the effects of other treatment and harm reduction modalities including slow-release oral morphine, injectable OAT (with diamorphine or hydromorphone), or the emerging practice prescribing a "safe supply" of pharmaceutical opioids to substitute for illicitly manufactured heroin or fentanyl [63].

## Conclusions

Among people with opioid use disorder following hospitalization for injecting-related bacterial or fungal infections, use of OAT is associated with lower risk of death or rehospitalization with

injecting-related infections. Our findings suggest that patients with opioid use disorder and injecting-related bacterial or fungal infections can reduce their risk of death or reinfection by engaging in OAT. Clinicians, hospitals, and health systems should facilitate access to OAT and support engagement.

## Supporting information

**S1 RECORD-PE Checklist. RECORD-PE, REporting of studies Conducted using Observational Routinely collected health Data statement for PharmacoEpidemiology.**
(PDF)

**S1 Table. ICD-10 codes to define infections of interest.**
(DOCX)

**S1 Fig. DAG describing hypothesized relationships between primary exposure, covariates, and outcomes.** Figure generated with Daggity.net software. Timing of variables generally goes from the left to right. Blue circle is outcome. Green circle is exposure. Red circles are ancestors of exposures and of outcomes. White circles are adjusted variables (in this case, through study design and selection criteria). Gray circles are unobserved variables (in this case, macroenvironmental influences on risk). DAG, directed acyclic graph.
(DOCX)

## Acknowledgments

Data were provided, and linkage was conducted by the NSW Ministry of Health, Centre for Health Record Linkage, and Bureau of Crime Statistics and Research. We also acknowledge the support and expertise of the OATS Study Aboriginal Advisory Group in reviewing this manuscript. The authors acknowledge the Registries of Births, Deaths and Marriages, the Coroners and the National Coronial Information System for enabling Cause of Death Unit Record File (COD URF) data to be used for this publication.

We thank the faculty and trainees in the Research in Addiction Medicine Scholars (RAMS) program and in the Fellows Immersion Training in Addiction Medicine (FIT) program for helpful feedback on the analysis plan.

## Disclaimers

The views expressed are those of the author(s) and not necessarily those of the NHS, the NIHR, or the Department of Health and Social Care.

## Author Contributions

**Conceptualization:** Thomas D. Brothers, Dan Lewer, Duncan Webster, Andrew Hayward, Louisa Degenhardt.

**Data curation:** Thomas D. Brothers, Dan Lewer, Nicola Jones, Samantha Colledge-Frisby, Louisa Degenhardt.

**Formal analysis:** Thomas D. Brothers, Dan Lewer.

**Investigation:** Thomas D. Brothers, Dan Lewer, Nicola Jones, Samantha Colledge-Frisby.

**Methodology:** Thomas D. Brothers, Dan Lewer, Nicola Jones, Samantha Colledge-Frisby, Michael Farrell, Matthew Hickman, Duncan Webster, Andrew Hayward, Louisa Degenhardt.

**Project administration:** Thomas D. Brothers, Nicola Jones, Michael Farrell.

**Resources:** Nicola Jones, Michael Farrell, Matthew Hickman, Louisa Degenhardt.

**Supervision:** Dan Lewer, Duncan Webster, Andrew Hayward, Louisa Degenhardt.

**Visualization:** Thomas D. Brothers, Dan Lewer.

**Writing – original draft:** Thomas D. Brothers.

**Writing – review & editing:** Thomas D. Brothers, Dan Lewer, Nicola Jones, Samantha Colledge-Frisby, Michael Farrell, Matthew Hickman, Duncan Webster, Andrew Hayward, Louisa Degenhardt.

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
