## [Editor Report · Decision Letter 0]

16 Feb 2022

Dear Dr Brothers, 

Thank you for submitting your manuscript entitled "Association of opioid agonist treatment with mortality and rehospitalization following injection drug use-associated bacterial and fungal infections: linkage cohort study" for consideration by PLOS Medicine.

Your manuscript has now been evaluated by the PLOS Medicine editorial staff and I am writing to let you know that we would like to send your submission for external assessment.

However, we first need you to complete your submission by providing the metadata that is required for full assessment. To this end, please login to Editorial Manager where you will find the paper in the 'Submissions Needing Revisions' folder on your homepage. Please click 'Revise Submission' from the Action Links and complete all additional questions in the submission questionnaire.

Please re-submit your manuscript within two working days, i.e. by Feb 18 2022 11:59PM.

Once your full submission is complete, your paper will undergo a series of checks in preparation for full assessment. 

Kind regards,

Richard Turner, PhD

rturner@plos.org

---

## [Decision Letter · Decision Letter 1]

20 Mar 2022

Dear Dr. Brothers,

Thank you very much for submitting your manuscript "Association of opioid agonist treatment with mortality and rehospitalization following injection drug use-associated bacterial and fungal infections: linkage cohort study" (PMEDICINE-D-22-00493R1) for consideration at PLOS Medicine. 

Your paper was discussed with an academic editor with relevant expertise and sent to independent reviewers, including a statistical reviewer. The reviews are appended at the bottom of this email and any accompanying reviewer attachments can be seen via the link below:

[LINK]

In light of these reviews, we will not be able to accept the manuscript for publication in the journal in its current form, but we would like to invite you to submit a revised version that addresses the reviewers' and editors' comments fully. You will appreciate that we cannot make a decision about publication until we have seen the revised manuscript and your response, and we expect to seek re-review by one or more of the reviewers. 

We hope to receive your revised manuscript by Apr 08 2022 11:59PM. Please email us (plosmedicine@plos.org) if you have any questions or concerns.

Please let me know if you have any questions, and we look forward to receiving your revised manuscript. 

Sincerely,

Richard Turner PhD

Senior editor, PLOS Medicine

rturner@plos.org

Please add "LD is a member of PLOS Medicine's Editorial Board", or similar, to the competing interest statement (submission form).

Please review PLOS' data policy (https://journals.plos.org/plosmedicine/s/data-availability) and adapt the data statement (submission form) to ensure that authors are not included as contacts for readers interested in inquiring about access to study data. 

We ask you to adapt the title to better match journal style, and suggest: "Opioid agonist treatment and mortality and rehospitalization following injection drug use-associated bacterial and fungal infections: A linkage cohort study".

Please add a new final sentence to the "Methods and findings" subsection of your abstract, beginning "Study limitations include ..." or similar and quoting 2-3 of the study's main limitations. 

After the abstract, please add a new and accessible "Author summary" section in non-identical prose. You may find it helpful to consult one or two recent research papers in PLOS Medicine to get a sense of the preferred style. 

At line 96 and any other instances, please substitute "sex" for "gender" as appropriate. 

Early in the Methods section (main text), please state whether or not the study had a protocol or prespecified analysis plan, and if so attach the document, referred to in the text. 

Please highlight analyses that were not prespecified. 

Throughout the text, please relocate reference call-outs to precede punctuation, e.g., "... similar pathways [1,25].".

Please remove the information on data sharing from the end of the main text. In the event of publication, this information will appear in the article metadata, via entries in the submission form. 

In the reference list, please convert all boldface text and italics to plain text. 

Please use the journal name abbreviations "PLoS ONE" and PLoS Med.". 

Where appropriate, please list 6 author names rather than 3, followed by "et al.".

Please add a completed checklist for the most appropriate reporting guideline, e.g., STROBE, as an attachment, labelled "S1_STROBE_Checklist" and referred to as such in the Methods section (main text). 

In the checklist, please refer to individual items by section (e.g., "Methods") and paragraph numbers, not by line or page numbers as these generally change in the event of publication. 

Comments from academic editor:

1. I see that time at risk begins immediately after discharge. Are "bounce backs" (i.e., readmission within a short time interval, e.g. 24 hours, 7 days, etc) therefore considered new hospitalizations? And if so, is there any reason to suspect this may bias the estimated associations away from the null? Perhaps a sensitivity analysis might be reassuring. Either way, please include some discussion of this in the text.

2. The authors state that the six days following an OAT episode is counted as part of the exposure. How was the decision made to specify the threshold at six days? Please describe in in the text and/or state whether or not a sensitivity analysis (e.g., with alternative threshold periods) is necessary.

3. Relatedly, the authors describe how person-time exposure was counted as "while exposed to OAT" vs. "while unexposed to OAT". Is there any need to account for any lag effects? For instance, suppose a study participant is unexposed to OAT and during this time engages in injection drug use. Then they re-engage in OAT and are therefore counted as exposed. But then 2 days later, due to the injection drug use behavior from several days before, this individual becomes sicker and requires hospitalization. They would be hospitalized but would be counted as exposed.

4. I would like to see the text provide more speculation about the magnitude of the difference between the effect of exposure on mortality vs. the effect of the exposure on rehospitalization. The large difference, and the fact that the mortality benefit occurs instantaneously (i.e., after discharge), makes me worry about potentially unobserved confounding.

5. Was the effect of exposure similar for Aboriginal or Torres Strait Islanders vs. non-Aboriginal or Torres Strait Islanders?"

Comments from the reviewers:

*** Reviewer #1: 

I mostly confine my remarks to statistical aspects of this paper. These were generally very well done, but I do have a few comments and suggestions.

p. 2 line 32 What is the number after the ± sign? An sd? A CI? Something else? Please specify.

p. 4 line 48-49 "Many parts of the world" ought to include some places in Africa, Asia, and less developed countries generally. If there's no data on those countries then maybe say "in many developed countries" or something like that.

p. 6-7 line 104-5 Maybe I am missing something but are the "eligible index hospitalizations" the control group? The linking mechanism is described earlier, but then this group doesn't seem to be described in the section on particiapants.

p. 8 line 139 Centering age is fine, but I am not a fan of scaling variables to 1 SD. I know some statisticians do recommend it, but I think it makes things a little less clear. For age (as here) a year is a year, but an SD in one study is different from an SD in another study. I won't insist on changing this, but I do recommend using age. (Same comment for LOS, below).

p, 9 line 148-9 Why group admission? Categorizing continuous variables is not recommended. It increases both type 1 and type 2 error and introduces a kind of "magical thinking" that something big happens at the cutpoints. Instead, leave year as a number and use splines to investigate nonlinearity. (Splines could also be used with age). 

Table 1 - LOS is clearly very skew. Giving median and IQR is fine, but a density plot of this variable would give more insight. What accounts for the huge SD? Is it a single outlier (someone with 500 days or something) or is it more general right skew? A density plot or quantile plot would reveal this.

Peter Flom

*** Reviewer #2: 

In this study, Brothers et all use data from the Australia OAT registry to example how the exposure of recent OAT relates to 2 outcomes: death and hospitalization. The authors are clearly very informed about the subject area, and I know their work well. My major issue with this is that the authors do not highlight as much why this specific registry is interesting and notable. There is a tension between "We know this" and "We don't know this." After reading this introduction, it is a hard sell to me that this work needs to be done. But, as you read the methods more, you realize how special this data set is because it is a closed system with really granular data. I say closed system not exactly knowing how many people leave New South Wales to go live in other places in Australia. Overall I think it is less than, say, people moving from Massachusetts to Rhode Island or New Hampshire. We would never be able to publish this type of data in the US. Additionally, 15% of people in this study are aboriginal. And I am not sure how comfortable the authors feel about diving into the details about disparities in OAT to aboriginal people, but this seems like the right time to use this data to make that statement. Yes, it will not work towards the idea of "generalizability" because not all readers will interact with aboriginal people. But it is hard to find this type of database with high percentages of native americans, and it could at least be the start to a conversation about why certain people are left out of the conversation. It looks like in the adjusted analysis the benfit of OAT goes away for both outcomes. So, I think this deserves to be published but the authors need to work on convincing me to read this paper. And the angle to sell this paper is the unique data set and the health disparities angle for sure. I recommend reframing the paper. Benefits of OAT in PWID Not Seen In Aboriginal People: Why? 

Introduction:

Concise and overall well done. In the first paragraph I might say specifically what the primary and secondary prevention methods are (safe injection sites, early identification of OUD, decreased barriers to OAT access). On the revisions it makes sense to reframe talking about major issues with known data that this paper/this data set can help answer. If you look at work by leaders like Laura Marks or Sim Kimmel, I think in the have people who are "lost to follow up" or they discuss issues with the linkage of data. I believe it has been part of my work as well, because I can only tell if people die in MA. I also don't think you can say OAT has not been prioritized without saying there are some places who have prioritized OAT, but many places have not. Do you think there is limited evidence that OAT prevents against death and rehospitalization? I think that is a stretch. I think there is more and more. And maybe that is where my disconnect lies with the paper. I think there is a lot of evidence about OAT protecting against hospitalization and death. I like your comment about limited sample size. It gives you another avenue to say this is a large group of people. Also please highlight more that this cohort includes incarcerated people (or, is it history of incarceration?). That is an important point and distinction

Methods:

-What is an unplanned hospitalization? You mean if they were delivering a baby, and happened to note there was cellulitis, they did not get included? If they had a planned hospitalization and found to have some sort of infection, why would they not be included? Maybe there is a better word for "unplanned" that can help me understand? I am curious about the 776 planned hospitalizations. I am guessing they were planned AND there was no infection? 

-I understand why you are selecting only outpatient OAT, but I think you need to be clearer in the intro about this. Also, if someone is going to a rehab on OAT, I would think they are sicker and more likely to be rehospitalizated. That is 1/10th of your sample (983 people). Is it possible to look at rehospitalization and death in the 983 people who did not go to community? Can you include this in your conclusions? You are selecting a healthier population over all.

-mycobaterial infections excluded? We have peopel with injection related mycobacterium. 

-Was the primary exposure before hospitalization, during hospitalization, or could be after hospitalization? Did they meet the primary exposure if they got just 1 day of meds?  (notes from later. Ok, I see in the table it is y/n prescription active at discharge; why do you need time varying based on day of receipt? not sure what that means.)

-Is it possible to go one step back from the 40,000 people and let me know how many people were in the OATs cohort all together onFigure 1

-History of incarceration may be linked to high risk injection practices, but I think current incarceration less so. I just think you need to justify/clarify this statement more. 

-I might be confused because I don't know Poisson well, but if Mr Jones is on and off OAT, does the days he is on OAT qualify for one of the exposures and the days he is off qualify him for the other? Or does he just fall into one bucket. 

Results

-I find the distribution of the infections fascinating. Such a high amount of hospitalizations for skin and soft tissue infections. Would be great if you could add somehting letting me know skin and soft tissue + something else. 

-lines 174 further confuses me. Is it TIME of index hospitalization? Or DISCHARGE? I need more clarity of the exposure of interest. I think I am confused becayse you are using it in 2 ways. Y/N, and then how much exposure for the KM curve. 

-Only 43% had prior experience of incarceration? How did you collect this data? Can you add into methods? Self report or pulled from admin data? 

-What do. you think about changing your KM curves into stratified analysis of Aboriginal and note. The Aboriginal peopel are not seeing the benefit. I feel like that is the biggest statemennt you can make. There is benefit but it is not equal. 

Discussion

-Rework discussion to be around equity. You found something, but it was not equitable benefit. That I believe is how this paper really makes changes and advances the discussion

-LIne 285-288. Not super relevant to your discussion. You excluded these people

-Discuss the other limitations, like excluding people who go to rehab from your analysis.

-Limiations of how places without high percentages of aboriginal people may not feel this applies, but it is about "MINORITIZED" communities overall, not necessarily WHY they are minoritized

*** Reviewer #3: 

This manuscript describes the association of opioid agonist treatment (methadone and buprenorphine) with mortality and re-hospitalization following injection drug use-associated bacterial and fungal infections in New South Wales, Australia between 2001 and 2018. The authors find that current OAT use is associated with both decreased rehospitalization and mortality in adjusted models. Hospitals, public health practitioners, infectious disease and addiction clinicians all seek to improve care for people with injection-related infections and this manuscript adds to the evidence. The manuscript is clear and well written and makes an important contribution to the field. I have several suggestions however for the authors to consider which I believe would improve the strength of this manuscript.

1. It seems that Individuals in the OATS cohort who experienced a designated hospitalization for infection were eligible for inclusion regardless of whether OAT was received prior to the hospitalization or after the hospitalization. Though this may be a small number of individuals, it is possible that someone could have received OAT only several years after experiencing a designated hospitalization and still be included in the cohort. This represents a form of immortalized time bias --- individuals had to have survived the period after the hospitalization in order to be included in the cohort. This potential bias however would be toward the null. To protect against biased estimates of effect, the authors could consider only including individuals with any OAT receipt prior to the hospitalization in the cohort. 

2. Figure 2 would be more accurate if it also designated OAT receipt. 

2. The study includes differential follow up time after the hospitalization. The analysis includes methods to address this but assumes that the impact of OAT is the same throughout the entire follow up period. Given that the relationship to the infection and the outcome varies over time (ie. infection may be less relevant to the outcomes years out), it would be instructive to report time varying hazards for example (ie. year 1, year 2-3, and year 4-6). 

3. Figure 3 would be improved with life tables. Also, the average follow up time is 6 years, but the figure only include 3 years of follow up? 

4. This may be beyond the scope of this study, but given the limited evidence in the literature, it may be instructive to also report how the association between OAT and mortality and rehospitalization varies by type of infection (e.g. skin and soft tissue vs more serious infections). Such an analysis would be instructive but could also represent its own manuscript in the future.

***

[LINK]

---

## [Decision Letter · Decision Letter 2]

5 Jun 2022

Dear Dr. Brothers,

Thank you very much for re-submitting your manuscript "Opioid agonist treatment and risk of death or rehospitalization following injection drug use-associated bacterial and fungal infections: a linkage cohort study" (PMEDICINE-D-22-00493R2) for consideration at PLOS Medicine.

I have discussed the paper with our academic editor and it was also seen again by three reviewers. I am pleased to tell you that, once the remaining editorial and production issues are fully dealt with, we expect to be able to accept the paper for publication in the journal.

[LINK]

Please let me know if you have any questions, and we look forward to receiving the revised manuscript.   

Sincerely,

Richard Turner, PhD

rturner@plos.org

Requests from Editors:

We ask you to remove or reword "... will be reviewed by the OATS investigator team" from the data statement. This seems to suggest that the author group can prevent data release to interested parties, which would not be consistent with PLOS' data policy (https://journals.plos.org/plosmedicine/s/data-availability).

We suggest incorporating "in Australia" in the full title. 

At line 34 and any similar instances, please make that "39 years". Incidentally, we notice that mean and median ages are both quoted at different points in the ms, and you may wish to make this consistent. 

Around line 41, please adapt the summary of limitations so that this extends to only the final sentence of the "Methods and findings" subsection of the abstract. 

Please avoid "reduced" and "decreased" (risk of death), e.g., at lines 48, 397-399 and 492, in favour of "lower", for example, to avoid implying causality. 

At line 145-146, please substitute the label for the attached checklist.

In the reference list, please abbreviate "Drug Alcohol Depend." consistently. 

Noting reference 9, please use the journal name abbreviation "PLoS ONE". 

Please remove "[Internet]" from references 46, 57 and any others. 

Is reference 61 missing the year of publication?

Comments from Reviewers:

*** Reviewer #1: 

The authors have addressed my concerns and I now recommend publication

*** Reviewer #2: 

The authors have done a great job with all of the reviewers comments. Without hesitation, I think this paper is worthy of publication!

*** Reviewer #3: 

I thank the authors for their thorough and thoughtful revision. 

My questions and concerns have been thoroughly addressed and I support publication of this important contribution!

***

[LINK]

---

## [Editor Report · Decision Letter 3]

12 Jun 2022

Dear Dr Brothers, 

On behalf of my colleagues and the Academic Editor, Dr Tsai, I am pleased to inform you that we have agreed to publish your manuscript "Opioid agonist treatment and risk of death or rehospitalization following injection drug use-associated bacterial and fungal infections: a cohort study in New South Wales, Australia" (PMEDICINE-D-22-00493R3) in PLOS Medicine.

PRESS

Sincerely, 

Richard Turner, PhD 

rturner@plos.org